# On Blackbox Backpropagation and Jacobian Sensing

**Krzysztof Choromanski**
Google Brain
New York, NY 10011
kchoro@google.com

**Vikas Sindhwani**
Google Brain
New York, NY 10011
sindhwani@google.com

## Abstract

From a small number of calls to a given "blackbox" on random input perturbations, we show how to efficiently recover its unknown Jacobian, or estimate the left action of its Jacobian on a given vector. Our methods are based on a novel combination of compressed sensing and graph coloring techniques, and provably exploit structural prior knowledge about the Jacobian such as sparsity and symmetry while being noise robust. We demonstrate efficient backpropagation through noisy blackbox layers in a deep neural net, improved data-efficiency in the task of linearizing the dynamics of a rigid body system, and the generic ability to handle a rich class of input-output dependency structures in Jacobian estimation problems.

## 1 Introduction

Automatic Differentiation (AD) [1, 17] techniques are at the heart of several "end-to-end" machine learning frameworks such as TensorFlow [5] and Torch [2]. Such frameworks are organized around a library of primitive operators which are differentiable vector-valued functions of data inputs and model parameters. A composition of these primitives defines a *computation graph* - a directed acyclic graph whose nodes are operators and whose edges represent dataflows, typically culminating in the evaluation of a scalar-valued loss function. For reverse mode automatic differentiation (backpropagation) to work, each operator needs to be paired with a gradient routine which maps gradients of the loss function with respect to the outputs of the operator, to gradients with respect to its inputs. In this paper, we are concerned with extending the automatic differentiation paradigm to computation graphs where some nodes are "blackboxes" [12], that is, opaque pieces of code implemented outside the AD framework providing access to an operator only via expensive and potentially noisy function evaluation, with no associated gradient routine available. A useful mental model of this setting is shown below where $f_3$ is a blackbox.

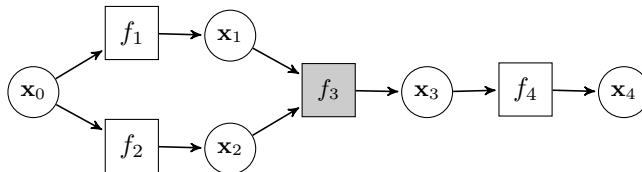

Blackboxes, of course, are pervasive - as legacy or proprietary codes or executables, numerical optimization routines, physics engines (e.g, Bullet [3] and MuJoCo [4]), or even wrappers interfacing with a mechanical system as is typically the case in reinforcement learning, robotics and process control applications.

The unknown Jacobian of a blackbox is the central object of study in this paper. Recall that the Jacobian $\nabla f(\mathbf{x}_0)$ of a differentiable vector-valued map $f : \mathbb{R}^n \mapsto \mathbb{R}^m$ at an input $\mathbf{x}_0 \in \mathbb{R}^n$ is the

$m \times n$ matrix of partial derivatives, defined by,

$$[\nabla f(\mathbf{x}_0)]_{ij} = \frac{\partial f_i}{\partial x_j}(\mathbf{x}_0)$$

The rows of the Jacobian are gradient vectors of the $m$ component functions $f = (f_1 \ldots f_m)$ and the columns are indexed by the $n$-dimensional inputs $\mathbf{x} = (x_1 \ldots x_n)$. Through Taylor approximation, the Jacobian characterizes the rate of change in $f$ at a step $\epsilon$ ($0 < \epsilon \ll 1$) along any direction $\mathbf{d} \in \mathbb{R}^n$ in the neighborhood of $\mathbf{x}_0$ as follows,

$$\nabla f(\mathbf{x}_0)\mathbf{d} \approx \frac{1}{\epsilon}\left[f(\mathbf{x}_0 + \epsilon\mathbf{d}) - f(\mathbf{x}_0)\right]. \tag{1}$$

Viewed as a linear operator over perturbation directions $\mathbf{d} \in \mathbb{R}^n$, differences of the form $\frac{1}{\epsilon}[f(\mathbf{x} + \epsilon\mathbf{d}) - f(\mathbf{x})]$ may be interpreted as noisy measurements ("sensing" [10, 11, 13]) of the Jacobian based on function evaluation. The measurement error grows with the step size $\epsilon$ and the degree of nonlineary in $f$ in the vicinity of $\mathbf{x}_0$. Additional measurement noise may well be introduced by unknown error-inducing elements inside the blackbox.

From as few perturbations and measurements as possible, we are concerned with approximately recovering either the full Jacobian, or approximating the action of the *transpose* of the Jacobian on a given vector in the context of enabling backpropagation through blackbox nodes. To elaborate on the latter setting, let $\mathbf{y} = f(\mathbf{x})$ represent forward evaluation of an operator, and let $\mathbf{p} = \frac{\partial l}{\partial \mathbf{y}}$ be the gradient of a loss function $l(\cdot)$ flowing in from the "top" during the reverse sweep. We are interested in approximating $\frac{\partial l}{\partial \mathbf{x}} = [\nabla f(\mathbf{x})]^T \mathbf{p}$, i.e. the action of the transpose of the Jacobian on $\mathbf{p}$. Note that due to linearity of the derivative, this is the same as estimating the gradient of the scalar-valued function $g(\mathbf{x}) = \mathbf{p}^T f(\mathbf{x})$ based on scalar measurements of the form $\frac{1}{\epsilon}\left(g(\mathbf{x} + \epsilon\mathbf{d}) - g(\mathbf{x})\right)$, which is a special case of the tools developed in this paper.

The more general problem of full Jacobian estimation arises in many derivative-free optimization settings [12, 8]. Problems in optimal control and reinforcement learning [18, 21, 20] are prominent examples, where the dynamics of a nonlinear system (e.g., a robot agent) needs to be linearized along a trajectory of states and control inputs reducing the problem to a sequence of time-varying Linear Quadratic Regulator (LQR) subproblems [21]. The blackbox in this case is either a physics simulator or actual hardware. The choice of perturbation directions and the collection of measurements then becomes intimately tied to the agent's strategy for exploration and experience gathering.

Finite differencing, where the perturbation directions $\mathbf{d}$ are the $n$ standard basis vectors, is a default approach for Jacobian estimation. However, it requires $n$ function evaluations which may be prohibitively expensive for large $n$. Another natural approach, when the number of measurements, say $k$, is smaller than $n$, is to estimate the Jacobian via linear regression,

$$\underset{\mathbf{J} \in \mathbb{R}^{m \times n}}{\operatorname{argmin}} \sum_{i=1}^{k} \|\mathbf{J}\mathbf{d}^i - \frac{1}{\epsilon}\left[f(\mathbf{x}_0 + \epsilon\mathbf{d}^i) - f(\mathbf{x}_0)\right]\|_2^2 + \lambda\|\mathbf{J}\|_F^2,$$

where an $l_2$ regularizer is added to handle the underdetermined setting and $\|\cdot\|_F$ stands for the Frobenius norm. This approach assumes that the error distribution is Gaussian and in its basic form, does not exploit additional Jacobian structure, e.g., symmetry and sparsity, to improve data efficiency. For example, if backpropagation needs to be enabled for a noiseless blackbox with identical input-output dimensions whose unknown Jacobian happens to be symmetric, then just one function evaluation suffices since $\nabla f(\mathbf{x}_0)^T \mathbf{p} = \nabla f(\mathbf{x}_0)\mathbf{p} \approx \frac{1}{\epsilon}(f(\mathbf{x}_0 + \epsilon\mathbf{p}) - f(\mathbf{x}_0))$. Figure 1 shows the histogram of the Jacobian of the dynamics of a Humanoid walker with respect to its 18-dimensional state variables and 6 dimensional control inputs. It can be seen that the Jacobian is well approximated by a sparse matrix. In a complex dynamical system comprising of many subsystems, most state or control variables only have local influence on the instantaneous evolution of the overall state. Figure 1 also shows the example of a manipulator; the Jacobian of a 5 planar link system has sparse and symmetric blocks (highlighted by blue and red bounding boxes) as a consequence of the form of the equations of motion of a kinematic tree of rigid bodies. Clearly, one can hope that incorporating this kind of prior knowledge in the Jacobian estimation process will improve data efficiency in "model-free" trajectory optimization applications.

**Technical Preview, Contributions and Outline**: We highlight the following contributions:

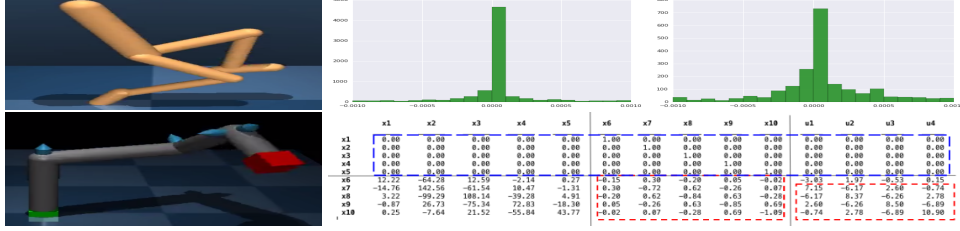

Figure 1: Structured Jacobians in Continuous Control Problems

- In §2: We start by asking how many blackbox calls are required to estimate a sparse Jacobian with known sparsity pattern. We recall results from automatic differentiation [14, 17, 23] literature that relates this problem to graph coloring [19, 26] where the chromatic number of a certain graph that encodes input-output dependencies dictates the sample complexity. We believe that this connection is not particularly well known in the deep learning community, though coloring approaches only apply to noiseless structure-aware cases.
- In §3: We present a Jacobian recovery algorithm, `rainbow`, that uses a novel probabilistic graph coloring subroutine to reduce the effective number of variables, leading to a compressed convex optimization problem whose solution yields an approximate Jacobian. The approximation $\widehat{\mathbf{J}}$ of the true Jacobian $\mathbf{J}$ is such that $\|\widehat{\mathbf{J}} - \mathbf{J}\|_F \leq E(n)$, where the measurement error vector $\boldsymbol{\eta} \in \mathbf{R}^m$ satisfies: $\|\boldsymbol{\eta}\|_\infty = o(E(n))$. Our algorithm requires only $O(\min(A, B))$ calls to the blackbox, where $A = d_{\text{int}} \log^2(\frac{\sqrt{mn}}{E(n)})$, $B = m\rho(\mathbf{J}, G_{\text{int}}^{\text{weak}}) \log^2(\frac{\sqrt{m\rho(\mathbf{J}, G_{\text{int}}^{\text{weak}})}}{E(n)})$, $d_{\text{int}}$ is a measure of intrinsic dimensionality of a convex set $\mathcal{C} \ni \mathbf{J}$ encoding prior knowledge about the Jacobian (elaborated below) and $\rho(\mathbf{J}, G_{\text{int}}^{\text{weak}}) \leq n$ is a parameter encoding combinatorial properties possibly known in advance (encoded by the introduced later the so-called *weak-intersection graph* $G_{\text{int}}^{\text{weak}}$) of the sparsity pattern in the Jacobian (see: §3.4.1 for an explicit definition); we will refer to $\rho(\mathbf{J}, G_{\text{int}}^{\text{weak}})$ as the *chromatic character of* $\mathbf{J}$.
- We demonstrate our tools with the following experiments: (1) Training a convolutional neural network in the presence of a blackbox node, (2) Estimating structured Jacobians from few calls to a blackbox with different kinds of local and global dependency structures between inputs and outputs, and (3) Estimating structured Jacobians of the dynamics of a 50-link manipulator, with a small number of measurements while exploiting sparsity and partial symmetry via priors in $l_p$ regression.

The convex set $\mathcal{C}$ mentioned above can be defined in many different ways depending on prior knowledge about the Jacobian (e.g., lower and upper bounds on certain entries, sparsity with unknown pattern, symmetric block structure, etc).

As we show in the experimental section, our approach can be applied also for non-smooth problems where Jacobian is not well-defined. Note that in this setting one can think about a nonsmooth function as a noisy version of its smooth approximation and a Jacobian of a function smoothing (such as Gaussian smoothing) is a subject of interest.

**Notation:** $\mathbf{D} = [\mathbf{d}_1 \ldots \mathbf{d}_k] \in \mathbb{R}^{n \times k}$ will denote the matrix of perturbation directions, with the corresponding measurement matrix $\mathbf{R} = [\mathbf{r}_1 \ldots \mathbf{r}_k] \in \mathbb{R}^{m \times k}$ where $\mathbf{r}_i = \frac{1}{\epsilon}[f(\mathbf{x} + \epsilon \mathbf{d}_i) - f(\mathbf{x})]$.

## 2 The Link between Jacobian Estimation and Graph Coloring

Suppose the Jacobian is known to be a diagonal matrix. Then finite differencing where perturbation directions are the $n$ standard basis elements is utterly wasteful; it is easy to see that a single perturbation direction $\mathbf{d} = [1, 1 \ldots 1]^T$ suffices in identifying all diagonal elements. The goal of this section is to explain the connection between Jacobian recovery and graph coloring problems that substantially generalizes this observation.

First we introduce graph theory terminology. The undirected graph is denoted as $G(V, E)$, where $V$ and $E$ stand for the sets of vertices and edges respectively. For $v, w \in V$ we say that $v$ is *adjacent to $w$* if there is an edge between $v$ and $w$. The degree $deg(v)$ of $v \in V$ is the number of vertices adjacent to it. The maximum degree in $G(V, E)$ will be denoted as $\Delta(G)$. A *stable set in $G$* is the

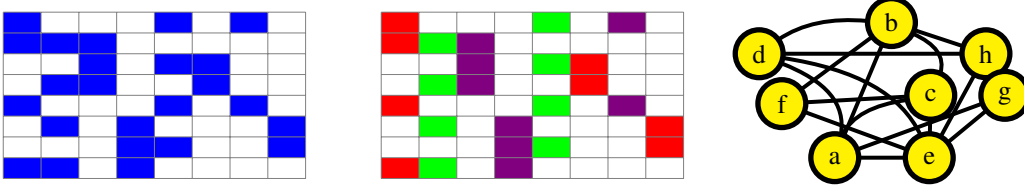

Figure 2: On the left: Sparse Jacobian for a function $f(a, b, c, d, e, f, g, h)$ with $n = m = 8$, where blue entries indicate nonzero values. In the middle: coloring of columns. A fixed color corresponds to a stable set in $G_{\text{int}}$. On the right: corresponding intersection graph $G_{\text{int}}$.

subset $S \subseteq V$, where no two vertices are adjacent. The *chromatic number* $\chi(G)$ *of* $G$ is the minimum number of sets in the partitioning of $V$ into stable sets. Equivalently, it is the smallest number of colors used in a valid vertex-coloring of the graph, where a valid coloring is one in which adjacent vertices are assigned different colors.

Denote by $\mathbf{J^x} = [\mathbf{J}^1, ..., \mathbf{J}^n] \in \mathbb{R}^{m \times n}$ a Jacobian matrix evaluated at point $\mathbf{x} \in \mathbb{R}^n$, where $\mathbf{J}^i \in \mathbb{R}^m$ denotes the $i$-th column. Assume that $\mathbf{J}^i$s are not known, but the sparsity structure, i.e. the location of zero entries in $\mathbf{J}$ is given. Let $A_i = \{k : J_k^i \neq 0\} \subseteq \{0, ..., m-1\}$ be the indices of the non-zero elements of $\mathbf{J}^i$. The *intersection graph*, denoted by $G_{\text{int}}$, is a graph whose vertex set is $V = \{x_1 \dots x_n\}$ and $x_i$ is adjacent to $x_j$ if the sets $A_i$ and $A_j$ intersect. In other words, there exists an output of the blackbox that depends both on $x_i$ and $x_j$ (see Figure 2 for an illustration). Now suppose $k$ colors are used in a valid coloring of $G_{\text{int}}$. The key fact that relates the Jacobian recovery problem to graph coloring is the following observation. If one constructs vectors $\mathbf{d}^i \in \mathbb{R}^n$ for $i = 1, ..., k$ in such a way that $d_j^i = 1$ if $x_j$ is colored by the $i^{th}$ color and is 0 otherwise, then $k$ computations of the finite difference $\frac{f(\mathbf{x} + \epsilon \mathbf{d}^i) - f(\mathbf{x})}{\epsilon}$ for $0 < \epsilon \ll 1$ and $i = 1, ..., k$ suffice to accurately approximate the Jacobian matrix (assuming no blackbox noise). The immediate corollary is the following lemma.

**Lemma 2.1** ([14]). *The number of calls $k$ to a blackbox vector-valued function $f$ needed to compute an approximate Jacobian via finite difference technique in the noiseless setting satisfies $k \leq \chi(G_{\text{int}})$, where $G_{\text{int}}$ is the corresponding intersection graph.*

Thus, blackboxes whose unknown Jacobian happens to be associated with intersection graphs of low chromatic number admit accurate Jacobian estimation with few function calls. Rich classes of graphs have low chromatic number. If the maximum degree $\Delta(G_{\text{int}})$ of $G_{\text{int}}$ is small then $\chi(G_{\text{int}})$ is also small, because of the well known fact that $\chi(G_{\text{int}}) \leq \Delta(G_{\text{int}}) + 1$. For instance if every input $x_i$ influences at most $k$ outputs $f_j$ and every output $f_j$ depends on at most $l$ variables $x_i$, then one can notice that $\Delta(G_{\text{int}}) \leq kl$ and thus $\chi(G_{\text{int}}) \leq kl + 1$. When the maximum degree is small, an efficient coloring can be easily found by the greedy procedure that colors vertices one by one and assigns to the newly seen vertex the smallest color that has not been used to color all its already seen neighbors ([14]). This procedure cannot be applied if there exist vertices of high degree. That is the case for instance if there exist few global variables influence a large number of outputs $f_i$. In the subsequent sections we will present an algorithm that does not need to rely on the small value of $\Delta(G_{\text{int}})$.

Graph coloring for Jacobian estimation has two disadvantages even if we assume that good quality coloring of the intersection graph can be found efficiently (optimal graph coloring is in general NP hard). It assumes that the sparsity structure of the Jacobian, i.e. the set of entries that are zero is given, and that all the measurements are accurate, i.e. there is no noise. We relax these limitations next.

## 3 Sensing and Recovery of Structured Jacobians

Our algorithm receives as input two potential sources of prior knowledge about the blackbox:

- sparsity pattern of the Jacobian in the form of a supergraph of the true intersection graph, which we call the *weak intersection graph* denoted as $G_{\text{int}}^{\text{weak}}$. The knowledge of the sparsity pattern may be imprecise in the sense that we can overestimate the set of outputs an input can influence. Note that any stable set of $G_{\text{int}}^{\text{weak}}$ is a stable set in $G_{\text{int}}$ and thus we have: $\chi(G_{\text{int}}) \leq \chi(G_{\text{int}}^{\text{weak}})$. A complete

weak intersection graph corresponds to the setting where no prior knowledge about the sparsity pattern is available while $G_{\text{int}}^{\text{weak}} = G_{\text{int}}$ reflects the setting with exact knowledge.

- a convex set $\mathcal{C}$ encoding additional information about the local and global behavior of the blackbox. For example, if output components $f_i$ are Lipschitz continuous with the Lipschitz constant $L_i$: the magnitude of the Jacobian entries can be bounded row-wise with $L_i, i = 1 \dots m$. The Jacobian may additionally have sparse blocks, which may be expressed as a bound on the elementwise $l_1$ norm over the entries of the block; it may also have symmetric and/or low-rank blocks [6] (the latter may be expressed as a bound on the nuclear norm of the block). A measure of the effective degrees of freedom due to such constraints directly shows up in our theoretical results on Jacobian recovery (§3.4).

Direct domain knowledge, or a few expensive finite-difference calls may be used in the first few iterations to collect input-independent structural information about the Jacobian, e.g., to observe the typical degree of sparsity, whether a symmetry or sparsity pattern holds across iterations etc.

Our algorithm, called `rainbow`, consists of three steps:

- `Color`: Efficient coloring of $G_{\text{int}}^{\text{weak}}$ for reducing the dimensionality of the problem, where each variable in the compressed problem corresponds to a subset of variables in the original problem. This phases explores strictly combinatorial structural properties of $\mathbf{J}$ (§3.1).
- `Optimize`: Solving a compressed convex optimization problem to minimize (or find a feasible) $l_p$ reconstruction. This phase can utilize additional structural knowledge via the convex set $\mathcal{C}$ ((§3.3)) defined earlier.
- `Reconstruct`: Mapping the auxiliary variables from the solution to the above convex problem back to the original variables to reconstruct $\mathbf{J}$.

Next we discuss all these steps.

### 3.1 Combinatorial Variable Compression via Graph Coloring: GreedyColoring

Consider the following coloring algorithm for reducing the effective number of input variables. Order the vertices $x_1, ..., x_n$ of $G_{\text{int}}^{\text{weak}}$ randomly. Initialize the list of stable sets $I$ covering $\{x_1, ..., x_n\}$ as $I = \emptyset$. Process vertices one after another and add a vertex $x_i$ to the first set from $I$ that does not contain vertices adjacent to $x_i$. If no such a set exists, add the singleton set $\{x_i\}$ to $I$. After processing all the vertices, each stable set from $I$ gets assigned a different color. We denote by $\text{color}(i)$ the color assigned to vertex $x_i$ and by $l$ the total number of colors. To boost the probability of finding a good coloring, one can repeat the procedure above for a few random permutations and choose the one that corresponds to the smallest $l$.

### 3.2 Choice of Perturbation Directions

Each $\mathbf{d}^i \in \mathbb{R}^n$ is obtained from the randomly chosen vector $\mathbf{d}_{\text{core}}^i \in \mathbb{R}^l$, that we call the *core vector*. Entries of all core vectors are taken independently from the same distribution $\phi$ which is: Gaussian, Poissonian or bounded and of nonzero variance (for the sake of readability, technical conditions and extensions to this family of distributions is relegated to the Appendix). Directions may even be chosen from columns of structured matrices, i.e., Circulant and Toeplitz [7, 24, 22, 16]. Each $\mathbf{d}^i$ is defined as follows: $d^i(j) = d_{\text{core}}^i(\text{color}(j))$.

### 3.3 Recovery via Compressed Convex Optimization

**Linear Programming**: Assume that the $l_p$-norm of the noise vector $\boldsymbol{\eta} \in \mathbb{R}^m$ is bounded by $\epsilon = E(n)$, where $E(\cdot)$ encodes non-decreasing dependence on $n$. With the matrix of perturbation vectors $\mathbf{D} \in \mathbb{R}^{n \times k}$ and a matrix of the corresponding core vectors $\mathbf{D}_{\text{core}} \in \mathbb{R}^{l \times k}$ in hand, we are looking for the solution $\mathbf{X} \in \mathbb{R}^{m \times l}$ to the following problem:

$$\|(\mathbf{X}\mathbf{D}_{\text{core}} - \mathbf{R})_i\|_p \leq \epsilon, \ \ i = 1 \dots k \tag{2}$$

where subscript $i$ runs over columns, $\mathbf{R} \in \mathbb{R}^{m \times k}$ is the measurement matrix for the matrix of perturbations $\mathbf{D}$. For $p \in \{1, \infty\}$, this task can be cast as a Linear Programming (LP) problem. Note that the smaller the number of colors, $l$, the smaller the size of the LP. If $\mathcal{C}$ is a polytope, it can be included as additional linear constraints in the LP. After solving for $\mathbf{X}$, we construct the Jacobian approximation $\widehat{\mathbf{J}}$ as follows: $\widehat{\mathbf{J}}_{u,j} = \mathbf{X}_{u,\text{color}(j)}$, where $\text{color}(j)$ is defined above.

We want to emphasize that a Linear Programming approach is just one instantiation of a more general method we present here. Below we show another one based on ADMM for structured $l_2$ regression.

**ADMM Solvers for multiple structures:** When the Jacobian is known to have multiple structures, e.g., it is sparse and has symmetric blocks, it is natural to solve structured $l_2$ regression problems of the form,

$$\operatorname*{argmin}_{\mathbf{X} \in \mathbb{R}^{m \times l} \in \mathcal{S}} \sum_{i=1}^{k} \|(\mathbf{X}\mathbf{D}_{core} - \mathbf{R})_i\|_2^2 + \lambda\|\mathbf{X}\|_1,$$

where the convex constraint set $\mathcal{S}$ is the set of all matrices conforming to a symmetry pattern on selected square blocks; an example is the Jacobian of the dynamics of a 5-link manipulator as shown in Figure 1. A consensus ADMM [9] solver can easily be implemented for such problems involving multiple structural priors and constraints admitting cheap proximal and projection operators. For the specific case of the above problem, it runs the following iterations:

- Solve for $\mathbf{X}_1$: $\mathbf{X}_1^T = [\mathbf{D}_{core}\mathbf{D}_{core}^T + \rho\mathbf{I}]^{-1}\left(\mathbf{D}\mathbf{R}^T + \rho(\mathbf{X}^T - \mathbf{U}_1^T)\right)$
- $\mathbf{X}_2 = \texttt{symmetrize}[\mathbf{X} - \mathbf{U}_2, \mathcal{S}]$
- $\mathbf{X} = \texttt{soft-threshold}[\frac{1}{2}(\mathbf{X}_1 + \mathbf{X}_2 + \mathbf{U}_1 + \mathbf{U}_2), \lambda\rho^{-1}]$
- $\mathbf{U}_i = \mathbf{U}_i + \mathbf{X}_i - \mathbf{X}, \quad i = 1, 2$

where $\mathbf{X}_1, \mathbf{X}_2$ are primal variables with associated dual variables $\mathbf{U}_1, \mathbf{U}_2$, $\rho$ is the ADMM step size parameter, and $\mathbf{X}$ is the global consensus variable. The $\texttt{symmetrize}(\mathbf{X}, \mathcal{S})$ routine implements exact projection onto symmetry constraints - it takes a square block $\hat{\mathbf{X}}$ of $\mathbf{X}$ specified by the constraint set $\mathcal{S}$ and symmetrizes it simply as $\frac{1}{2}[\hat{\mathbf{X}} + \hat{\mathbf{X}}^T]$ keeping other elements of $\mathbf{X}$ intact. The soft-thresholding operator is defined by $\texttt{soft-threshold}(\mathbf{X}, \lambda) = \max(\mathbf{X} - \lambda, 0) - \max(-\mathbf{X} - \lambda, 0)$. Note that for the first step $[\mathbf{D}_{core}\mathbf{D}_{core}^T + \rho\mathbf{I}]$ can be factorized upfront, even across multiple Jacobian estimation problems since it is input-independent. Also, notice that if the perturbation directions are structured, e.g., drawn from a Circulant or Toeplitz matrix, then the cost of this linear solve can be further reduced using specialized solvers [15]. As before, after solving for $\mathbf{X}$, we construct the Jacobian approximation $\widehat{\mathbf{J}}$ as follows: $\widehat{\mathbf{J}}_{u,j} = \mathbf{X}_{u,\mathrm{color}(j)}$.

### 3.4 Theoretical Guarantees

#### 3.4.1 Chromatic property of a graph

The probabilistic graph coloring algorithm GreedyColoring generates a coloring, where the number of colors is close to the *chromatic property* $\Lambda(G_{\mathrm{int}}^{\mathrm{weak}})$ of the graph $G_{\mathrm{int}}^{\mathrm{weak}}$ (see: proof of Lemma 3.1 in the Appendix). The chromatic property $\Lambda(G)$ of a graph $G$ is defined recursively as follows.

- $\Lambda(G_\emptyset) = 0$, where $G_\emptyset$ is an empty graph ($V = \emptyset$),
- for $G \neq G_\emptyset$, we have: $\Lambda(G) = 1 + \max_{S \subseteq V} \Lambda(G \backslash S)$ where $\max$ is taken over all subsets satisfying: $|S| = |V| - \lceil \sum_{v \in V} \frac{1}{1 + deg(v)} \rceil$ and $G \backslash S$ stands for the graph obtained from $G$ be deleting vertices from $S$.

Note that we are not aware of any closed-form expression for $\Lambda(G)$. We observe that there exists a subtle connection between the chromatic property of the graph $\Lambda(G)$ and its chromatic number.

**Lemma 3.1.** *The following is true for every graph $G$: $\chi(G) \leq \Lambda(G)$.*

The importance of the chromatic property lies in the fact that in practice for many graphs $G$ (especially sparse, but not necessarily of small maximum degree $\Delta(G)$) the chromatic property is close to the chromatic number. Thus, in practice, GreedyColoring finds a good quality coloring for a large class of weak-intersection graphs $G_{\mathrm{int}}^{\mathrm{weak}}$, efficiently utilizing partial knowledge about the sparsity structure.

The *chromatic character of the Jacobian* is defined as the chromatic property of its weak-intersection graph $\Lambda(G_{\mathrm{int}}^{\mathrm{weak}})$ and thus does not depend only on the Jacobian $\mathbf{J}$, but also on its "sparsity exposition" given by $G_{\mathrm{int}}^{\mathrm{weak}}$ and will be referred to as $\rho(\mathbf{J}, G_{\mathrm{int}}^{\mathrm{weak}})$.

#### 3.4.2 Accuracy of Jacobian Recovery with `rainbow`

We need the following notion of intrinsic dimensionality in $\mathbb{R}^{m \times n}$ as a metric space equipped with $\|\cdot\|_F$ norm.

**Definition 3.2** (intrinsic dimensionality). *For any point $\mathbf{X} \in \mathbb{R}^{m \times n}$ and any $r > 0$, let $\mathcal{B}(\mathbf{X}, r) = \{\mathbf{Y} : \|\mathbf{X} - \mathbf{Y}\|_F \leq r\}$ denote the closed ball of radius $r$ centered at $\mathbf{X}$. The intrinsic dimensionality of $S \subseteq \mathbb{R}^{m \times n}$ is the smallest integer $d$ such that for any ball $\mathcal{B}(\mathbf{X}, r) \subseteq \mathbb{R}^{m \times n}$, the set $\mathcal{B}(\mathbf{X}, r) \cap S$ can be covered by $2^d$ balls of radius $\frac{r}{2}$.*

We are ready to state our main theoretical result.

**Theorem 3.3.** *Consider the Jacobian matrix $\mathbf{J} \in \mathbb{R}^{m \times n}$. Assume that $\max |J_{i,j}| \leq C$ for some fixed $C > 0$ and $\mathbf{J} \in \mathcal{C}$, where $\mathcal{C} \subseteq \mathbb{R}^{m \times n}$ is a convex set defining certain structural properties of $\mathbf{J}$ (for instance $\mathcal{C}$ may be the set of matrices with block sparsity and symmetry patterns). Assume that the measurement error vector $\boldsymbol{\eta} \in \mathbb{R}^m$ satisfies: $\|\boldsymbol{\eta}\|_\infty = o(E(n))$ for some function $E(n)$. Then the approximation $\widehat{\mathbf{J}}$ of $\mathbf{J}$ satisfying $\|\widehat{\mathbf{J}} - \mathbf{J}\|_F \leq E(n)$ can be found with probability $p = 1 - \frac{1}{\mathrm{spoly}(n)}$ by applying* `rainbow` *algorithm with $k = O(\min(A, B))$ calls to the $f$ function, where $A = d_{\text{int}} \log^2(\frac{C\sqrt{mn}}{E(n)})$, $B = m\rho(\mathbf{J}, G_{\text{int}}^{\text{weak}}) \log^2(\frac{C\sqrt{m\rho(\mathbf{J}, G_{\text{int}}^{\text{weak}})}}{E(n)})$, $d_{\text{int}}$ stands for the intrinsic dimensionality of $\mathcal{C}$ and $\mathrm{spoly}(n)$ is a superpolynomial function of $n$.*

The proof is given in the Appendix. The result above is a characterization of the number of blackbox calls needed to recover the Jacobian, in terms of its intrinsic degrees of freedom, the dependency structure in the inputs and outputs and the noise introduced by higher order nonlinear terms and other sources of forward evaluation errors.

## 4 Experiments

**4.1. Sparse Jacobian Recovery**: We start with a controlled setting where we consider the vector-valued function, $f : \mathbb{R}^n \to \mathbb{R}^m$ of the following form:

$$f(x_1, ..., x_n) = (\sum_{i \in \mathcal{S}_1} \sin(x_i), ..., \sum_{i \in \mathcal{S}_m} \sin(x_i)), \tag{3}$$

where sets $\mathcal{S}_i$ for $i = 1, ...., m$ are chosen according to one of the following models. In the $p$-model each entry $i \in \{1, ..., n\}$ is added to each $\mathcal{S}_j$ independently and with the same probability $p$. In the $\alpha$-model entry $i$ is added to each $\mathcal{S}_j$ independently at random with probability $i^{-\alpha}$. We consider a Jacobian at point $\mathbf{x} \in \mathbb{R}^n$ drawn from the standard multivariate Gaussian distribution with entries taken from $\mathcal{N}(0, 1)$. Both the models enable us to precisely control the sparsity of the corresponding Jacobian which has an explicit analytic form. Furthermore, the latter generates Jacobians where the degrees of the corresponding intersection graphs have power-law type distribution with few "hubs" very well connected to other nodes and many nodes of small degree. That corresponds to the setting, where there exist few global variables that impact many output $f_i$s, any many local variables that only influence a few outputs. We run the LP variant of `rainbow` for the above models and summarize the results in the table below.

| model | $m$ | $n$ | sparsity | $\chi/\Delta$ | $\sigma$ | $k$ | rel.error |
|---|---|---|---|---|---|---|---|
| $p = 0.1$ | 30 | **60** | 0.91277 | 0.33 | 0.07 | **15** | 0.0632 |
| $p = 0.1$ | 40 | **70** | 0.90142 | 0.35 | 0.07 | **20** | 0.0802 |
| $p = 0.1$ | 50 | **80** | 0.90425 | 0.32 | 0.07 | **30** | 0.0751 |
| $p = 0.3$ | 30 | 60 | 0.6866 | 0.6833 | 0.07 | 45 | 0.0993 |
| $p = 0.3$ | 40 | 70 | 0.7096 | 0.6857 | 0.07 | 60 | 0.0589 |
| $p = 0.3$ | 50 | 80 | 0.702 | 0.8625 | 0.07 | 70 | 0.1287 |
| $\alpha = 0.5$ | 30 | 60 | 0.7927 | 0.3833 | 0.1 | 45 | 0.0351 |
| $\alpha = 0.5$ | 40 | 70 | 0.78785 | 0.4285 | 0.1 | 60 | 0.0491 |
| $\alpha = 0.5$ | 50 | 80 | 0.79225 | 0.475 | 0.1 | 70 | 0.0443 |
| $\alpha = 0.7$ | 30 | 60 | 0.85166 | **0.2777** | 0.1 | 40 | 0.0393 |
| $\alpha = 0.7$ | 40 | 70 | 0.87357 | **0.2537** | 0.1 | 55 | 0.0398 |
| $\alpha = 0.7$ | 50 | 80 | 0.86975 | **0.275** | 0.1 | 65 | 0.0326 |

Above, we measure recovery error in terms of the relative Frobenius distance between estimated Jacobian and true Jacobian, rel.error $= \frac{\|\widehat{\mathbf{J}} - \mathbf{J}\|_F}{\|\mathbf{J}\|_F}$. The standard deviation of each entry of the measurement noise vector is given by $\sigma$. We report in particular the fraction of zero entries in $\mathbf{J}$ (sparsity), the ratio of the number of colors found by our GreedyColoring algorithm and the

maximum degree of the graph ($\frac{\chi}{\Delta}$). We see that the coloring algorithm finds good quality coloring even in the "power-law" type setting where maximum degree $\Delta(G)$ is large. The quality of the coloring in turn leads to the reduction in the number of measurement vectors needed ($k$) to obtain an accurate Jacobian approximation (i.e., relative error $< 0.1$).

**4.2. Training Convolutional Neural Networks with Blackbox Nodes**: We introduce a blackbox layer between the convolutional layers and the fully connected layers of a standard MNIST convnet. The blackbox node is a standard ReLU layer that takes as input 32-dimensional vectors, $32 \times 32$-sized weight matrix and a bias vector of length 32, and outputs a 32 dimensional representation. The minibatch size is 16. We inject truncated Gaussian noise in the output of the layer and override its default gradient operator in TensorFlow with our LP-based `rainbow` procedure. We use Gaussian perturbation directions and sample measurements by forward evaluation calls to the TensorFlow Op inside our custom blackbox gradient operator. In Fig. 3 we study the evolution of training and validation error across SGD iterations. We see in Fig. 3 that even though for low noise regime the standard linear regression and finite differencing methods work quite well, when noise magnitude increases our blackbox backpropagation procedure `rainbow`-LP shows superior robustness - retaining a capacity to learn while the other methods degrade in terms of validation error. The rightmost subfigure reports validation error for our method with different numbers of Jacobian measurements at a high noise level (in this case, the other methods fail to learn and are not plotted).

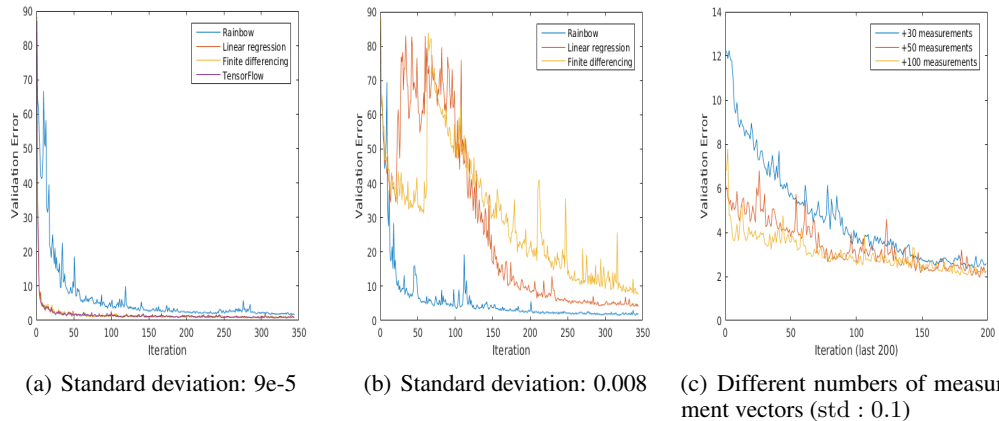

(a) Standard deviation: 9e-5  (b) Standard deviation: 0.008  (c) Different numbers of measurement vectors ($\mathrm{std}: 0.1$)

Figure 3: TensorFlow CNN training with a "blackbox" layer with `rainbow`-LP method. On the left: Comparison of `rainbow`-LP with finite differencing and linear regression methods for low noise regime. In the middle: As before, but for more substantial noise magnitude. On the right: `rainbow`-LP for even larger noise magnitude ($\mathrm{std}: 0.1$) and different number of measurement vectors used. In that setting other methods did not learn at all.

**4.3. Jacobian of manipulator dynamics**: We compute the true Jacobian of a planar rigid-body model with 50 links near an equilibrium point using MIT's Drake planning and control toolbox [25]. The first link is unactuated; the remaining are all torque-actuated. The state vector comprises of 50 joint angles and associated joint velocities, and there are 49 control inputs to the actuators. The Jacobian has sparse and symmetric blocks similar to Figure 1. We compare linear regression with $l_2$ regularization against the `rainbow` ADMM solver designed to exploit sparsity and symmetry, in the setting where the number of measurements is much smaller than the total number of input variables to the forward dynamics function (149). Results are shown in the adjacent Figure. The recovery is much more accurate in the presence of sparsity and symmetry priors. The results are similar if the matrix of perturbation directions are chosen from a Circulant matrix.

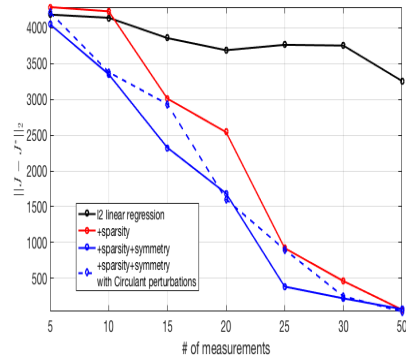

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
