[Reviews · NeurIPS 2017]

Reviewer 1



The paper focuses on automatic differentiation of multiple-variable vector-valued functioned in the context of training model parameters from input data. A standard approach in a number of popular environments rely on a propagation of the errors in the evaluation of the model parameters so far using backpropagation of the estimation error as computed of the (local) gradient of the loss function. In this paper, the emphasis is on settings where some of the operators of the model are exogenous blackboxes for which the gradient cannot be computed explicitly and one resorts to finite differencing of the function of interest. Such approach can be prohibitively expensive if the Jacobian does not have some special structure that can be exploited. The strategy pursued in this paper consists in exploiting the relationship between graph colouring and Jacobian estimation. In a nutshell, the mapping encodes the outputs that jointly influence the model. The consequence of this is a known result stating that the chromatic number of the associated graph provides a bound on the number of calls to a function to compute its Jacobian, in the noiseless setting. The paper goes one step beyond this established fact and proposes an algorithms “rainbow” that defines the perturbation directions where the blackbox needs to be evaluated in order to compute the associated Jacobian . The crux of the algorithm is to encode the properties of the Jacobian (e.g. sparsity, symmetry) using a graph colouring procedure, arising from the aforementioned graph mapping, into a simple linear programme. The authors theoretically evalute the performance of this algorithm in section 3.4. Theorem 3.3. upperbounds the number of blackbox needed to evaluate its Jacobian in terms of the graph encoding input output dependencies as well as its intrinsic degrees of freedom among other things. In Section 4, the proposed algorithm is evaluated in following settings: two synthetic input-output dependency structures (random and power-law), a CNN, and a planning and control setting. The paper is clearly written and properly spells out the relevant technical details of the proposed algorithm as well as its performance. The main weakness of the paper resides in section 3.3, focusing on the LP approach to define the perturbation direction needed to evaluate the Jacobian. There are also two minor points that would improve the legibility of the paper. First, the paragraph after theorem 3.3 (page 7 l.257-l .260) should provide further intuition about the bounds on the calls to the function needed by the algorithm to estimate its Jacobian. Then, the experiment section 4 would benefit from providing a rationale behind the choice of the scenarios explored and what is illustrated about the algorithm in each of them.

Reviewer 2



This is an interesting paper that discusses the problem of how to recover the Jacobian. By observing an interesting relationship between the number of needed measurements and the chromatic number of the so-called intersection graph, the paper found that the number of measurements can be significantly reduced if the chromatic number is small. Recovery guarantee is provided in the paper. As far as the reviewer can tell, the paper is technically correct. This paper has some novelty and is well written, thus is recommended to be accepted. I. The illustrated example is controversial. For such kind of systems, usually the system designer knows (from the physical model) that the Jacobians of some subsystems are zeros or identities, or of some specific structures. Thus, those parts (sub-matrices) of Jacobian should be determined directly and will not enter the measurement equations. In this case, for the current example, the remaining part of the Jacobian would not be sparse any more and would not seem to have any special structure. Can the authors show some real example in which the Jacobian is indeed sparse but the structures (special patterns) can not be physically determined? II. Problem (2) is actually a feasibility problem and should be easily handled with all p\geq 1 (convexity). III. More discussion on the bounds of k (chromatic number) is needed. After rebuttal: First of all, it seems the reviewer's questions are overlooked by the authors. Furthermore, the reviewer thinks the authors have over claimed'' a bit in the response to another reviewer. It is questionable if the method could work under non-smooth step-up. It is even unclear if the algorithm is still numerically stable when the problem becomes stiff. Overall speaking, it is a good work but the method's scope of applicability needs more responsible discussion.

Reviewer 3



The authors propose a Jacobi estimation method that could be used as a tool in back-propagation of neural networks with black-box units. This is definitely an interesting problem with practical implications. The theoretical results presented in the paper seem to be technically right. But the novelty of the results, in the context of automatic differentiation, is not very clear. Unfortunately, the authors do not provide any information on the novelty or importance of their theoretical results compared to the rest of the results in the automatic differentiation literature. There are also some other unanswered questions that require clarification. First of all, the algorithm requires knowing some underlying relationship graph or a good approximation of it. The number of samples that are required heavily depends on such graph, It is not clear to me how such graph could be obtained in practice when dealing with practical black boxes. Moreover, the Jacobian is defined for smooth functions. And in many cases, the black boxes seem to be non-smooth mappings. For example, the ReLU unit, that is used in experiments, is non-smooth. Therefore, the Jacobian is not defined for it. Although, there are some other concepts like approximate Jacobians that could be defined for non-smooth mappings. But the authors do not provide any information on if their algorithm is still theoretically/practically valid when applied to non-smooth mappings and what does it recover. In general, another question that should be addressed about the algorithm is how to select the size of perturbations in practice. Although having smaller perturbations is better in terms of approximating the Jacobian, it might result in very noisy or information-less observations if the size of perturbations is too small. For example, the mechanical system might not be even sensitive to very small perturbations. Moreover, the size of the perturbation can depend on the curvature of the mapping as well.